# Integration of Antifouling and Anti-Cavitation Coatings on Propellers: A Review

**Jingying Zhang [1,2,†], Weihua Qin [1,2,†], Wenrui Chen [1,2], Zenghui Feng [1,2], Dongheng Wu [1,2], Lanxuan Liu [1,2] and Yang Wang [1,2,\*]**

1   Wuhan Research Institute of Materials Protection, Wuhan 430030, China; zhangjy288@163.com (J.Z.); qinweihua0@163.com (W.Q.); andrewcwr@whut.edu.cn (W.C.); 13419694843@163.com (Z.F.); 18804050206@163.com (D.W.); liulanxuan@rimp.com.cn (L.L.)
2   State Key Laboratory of Special Surface Protection Materials and Application Technology, Wuhan 430030, China
\*   Correspondence: wangyang@rimp.com.cn
†   These authors contributed equally to this work.

**Abstract:** The performance of an entire ship is increasingly impacted by propellers, which are the essential components of a ship's propulsion system that have growing significance in a variety of aspects. Consequently, it has been a hot research topic and a challenge to develop high-performance antifouling and anti-cavitation coatings due to the issue of marine biofouling and cavitation faced by propellers in high-intensity service. While there is an overwhelming number of publications on antifouling and anti-cavitation coatings, a limited number of papers focus on integrated protective coatings on propellers. In this paper, we evaluated the development of antifouling and anti-cavitation coatings for ship propellers in the marine environment as well as their current status of research. These coatings include self-polishing antifouling coatings, fouling-releasing antifouling coatings, and biomimetic antifouling coatings for static seawater anti-biofouling, as well as anti-cavitation organic coatings and anti-cavitation inorganic coatings for dynamic seawater anti-cavitation. This review also focuses both on the domestic and international research progress status of integrated antifouling and anti-cavitation coatings for propellers. It also provides research directions for the future development of integrated antifouling and anti-cavitation coatings on propellers.

**Keywords:** propeller; antifouling coatings; anti-cavitation coatings; integration technology

## 1. Introduction

Propellers have been around for more than two hundred years, i.e., since the 19th century. Propellers have consistently been the preferred option for ship propulsion due to their notable efficiency and commendable hydrodynamic characteristics. Consequently, the overall performance and efficiency of an entire vessel are inherently influenced by the state of the propellers [1]. The presence of microorganisms and proteins in stagnant seawater poses a significant challenge in the maritime industry, as it leads to the attachment of marine organisms onto propellers and other materials, resulting in biofouling issues [2]. Simultaneously, the prolonged and rapid rotation of the propeller induces the development of cavitation on the material surface, leading to fatigue-related degradation of the propeller blades. Both of these factors have emerged as the primary determinants influencing the ship's performance, as depicted in Figure 1. The ship's propulsion system is expected to experience significant degradation due to the presence of biofouling and cavitation. These factors will lead to changes in the propeller's surface morphology, disrupt its dynamic balance, and result in reduced efficiency and range, accompanied by increased energy consumption. At present, the effectiveness of protective coatings, both within national borders and beyond international contexts, is below the desired standard. The significant challenges of seawater scouring and cavitation present considerable problems in the pursuit

of long-lasting protection, rendering it susceptible to succumbing to these deleterious effects. Therefore, an urgent need for the creation of a comprehensive protective coating for ship propellers that demonstrates resistance to both dynamic cavitation and static fouling has arisen.

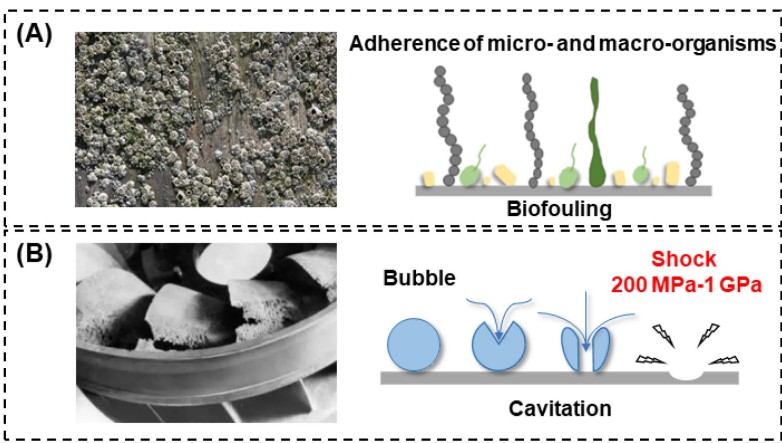

**Figure 1.** Ship propeller biofouling and cavitation erosion phenomenon and its mechanism diagram. (**A**) The phenomenon of marine biofouling and its mechanism; (**B**) The phenomenon of cavitation erosion and its mechanisms.

Marine biofouling refers to the phenomenon wherein submerged objects, such as marine propellers, pumps, turbines, and various components of the hull located beneath the waterline, become encrusted with biological organisms [3]. The attachment, propagation, and accumulation of marine organisms on the surface of these objects result in surface corrosion and material degradation [4]. The organisms responsible for this occurrence are referred to as marine fouling organisms, which can result in economic losses for human activities and livelihoods [5]. Additionally, this can result in unavoidable economic disadvantages in terms of human productivity and livelihood.

Based on statistical data, the global count of marine fouling organisms is estimated to range between 4000 and 5000 species [6]. Among these organisms, barnacles, mussels, oysters, and other similar species have been identified as the most detrimental along the coastal regions of China. The presence of marine fouling organisms has been found to have a direct impact on ship navigation by increasing the roughness of ship hull surfaces [7]. The presence of these substances has the potential to harm the protective coating designed to prevent corrosion on the hull, expedite the corrosion process on the underlying metal surface, escalate maintenance expenses, and result in significant financial setbacks. According to reports, the aggregate economic burden resulting from marine fouling organisms continues to surpass a staggering sum of 150 billion dollars annually. The proliferation and attachment of fouling organisms can result in an increase in weight and a decrease in the speed of a ship's hull [8]. Moreover, the presence of microorganisms adhered to ships can potentially exert an influence on the surrounding ecosystem [9].

The intricate process of marine biofouling has been found to be a dynamic and continuous phenomenon that can be divided into four distinct phases, as illustrated in Figure 2. A wide variety of organic components, including protein molecules, polysaccharides, and esters, immediately cling to the surface of the material in the early phase after burying it in saltwater. This phenomenon is attributed to several physical mechanisms, such as Brownian motion, van der Waals forces, electrostatic forces, and hydrogen bonding. Consequently, a provisional film is formed [10,11]. The subsequent phase involves the prompt adherence of microorganisms, specifically bacteria, facilitated by the adsorption of the conditioned film onto the metal substrate's surface. This process leads to the development of a biofilm, characterized by the production of metabolic secretions by the microorganisms and the entrapment of the polymer material itself [12,13]. In the third phase, the organisms responsible for pollution release various substances such as proteins, polysaccharides,

nucleic acids, and other compounds. These materials serve the dual purpose of capturing additional polluting species and providing them with essential nutrients to facilitate their growth. The fourth stage is characterized by the attachment and accumulation of minute organisms, leading to the settlement and growth of larvae from larger marine invertebrates and other organisms on the surface, ultimately resulting in substantial biofouling. The fouling process described above is generally applicable to the majority of marine organisms responsible for fouling in contemporary times, although it is not universally consistent. For instance, certain types of algae spores and barnacle larvae have the ability to directly adhere to material surfaces without the need for biofilms [14]. Furthermore, the marine environment exhibits a diverse array of marine fouling organisms, alongside its inherent complexity and variability. Several factors within the marine environment can affect the adhesion of marine fouling organisms to propeller surfaces. These factors include temperature, current velocity, shear stress, pH of seawater, and salinity. Each of these factors exerts a distinct influence on the fouling phenomenon that manifests on the surface of propellers.

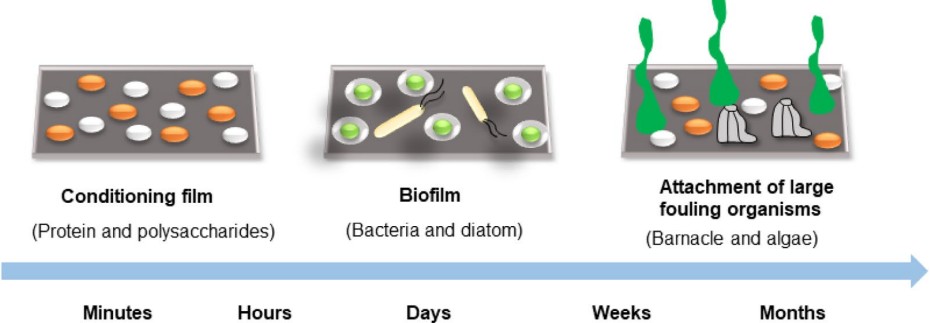

**Figure 2.** Diagram illustrating the marine biofouling process [15].

Cavitation is a unique form of erosion–corrosion that almost every propeller suffers from. The phenomenon of cavitation was initially observed and postulated in 1887 by the esteemed British scientist S.W. Barnaby during an investigation into the propeller efficiency of maritime vessels [16]. Currently, it is widely accepted that there are two factors contributing to the material degradation resulting from cavitation, as indicated in [17]. The initial cause can be attributed to the mechanical force that is generated upon the bursting of the bubble. The impact pressure wave generated by the collapse of the bubble at its center induces surface alterations in the nearby materials, leading to consequential material degradation [18]. On the other hand, the bubble undergoes collapse when subjected to deformation, and this deformation intensifies as the pressure increases, thereby facilitating the formation of a high-velocity microjet. These repeated impacts can lead to local plastic deformation of the materials and mass removal [19]. Furthermore, as a consequence of the influence of thermodynamics, when the bubble undergoes collapse within a region of elevated pressure, the vapor contained within the bubble, characterized by low pressure, swiftly condenses. This condensation process results in the liberation of a substantial quantity of heat, thereby inducing thermal damage of high temperature to the material [20]. However, propeller cavitation in a real marine environment involves the cooperative effects of mechanical-related corrosion and electrochemical corrosion, which act synergistically to accelerate the corrosion processes and therefore can be called erosion–corrosion. Cavitation is a prevalent and significant concern in hydraulic systems, encompassing various components such as turbine blades, valves, propellers, pipelines, and others. This phenomenon poses substantial financial losses and safety risks within industries such as ship water conservancy [21]. Based on the findings of the British Ship Research Association (BSRA), it has been observed that despite the relatively small surface area and volume of the propeller, the combined effects of biofouling and cavitation contribute to nearly one-third of the total loss [22]. Consequently, it is crucial to the utilization of integrated technology for the purpose of propeller surface antifouling and anti-cavitation.

## 2. Antifouling Coatings

The phenomenon of hull biofouling is characterized by its intricate and multifaceted nature. Various aspects such as different sections of the hull, depth of immersion, speed of the current, and additional factors play a significant role in determining the attachment behavior of marine fouling organisms. Organisms exhibiting robust adhesion tend to cause fouling on propellers as a consequence of prolonged exposure to seawater and the rapid rotational motion they experience. In a broad sense, fouling is primarily affected by the surface properties of the material, including the substrate's surface energy, wettability, and microstructure, among other factors. One of the most prevalent and efficacious methods for accomplishing this objective involves the utilization of antifouling coatings. Consequently, the deliberate alteration of the surface structure represents a more straightforward approach to managing fouling. Based on the antifouling properties exhibited by coatings, it is possible to categorize common antifouling coatings into three main types: self-polishing antifouling coatings, fouling release antifouling coatings, and bionic antifouling coatings. The application of antifouling coatings on propellers has the potential to mitigate energy dissipation, minimize frictional resistance between the propeller surface and saltwater, and restrict the attachment of marine fouling organisms [23]. As a result, the propellers of the ship will exhibit enhanced operational efficiency and durability, thereby more effectively fulfilling diverse requirements and minimizing financial losses.

### 2.1. Self-Polishing Antifouling Coatings

Self-polishing antifouling coatings function through the gradually hydrolyzing polymers, resulting in the formation of a peel layer on the external surface of the coating. The presence of this layer serves as a deterrent for marine fouling organisms, impeding their ability to adhere to surfaces such as ship propellers [24,25]. During the early 1970s, the development of organotin self-polishing antifouling coatings took place, utilizing analogs of butyltin (TBT) [26]. The primary function of this coating principle is to apply a coating onto a resin material based on acrylate, facilitating the hydrolysis of ester bonds and subsequently releasing the antifouling effect of TBT. Figure 3A provides an illustration of this technique. Simultaneously, the incorporation of an antifouling agent, such as cuprous oxide, within the paint film enables the release of copper ions. The combined effect of these ions enhances the coating's ability to resist fouling from a wide range of organisms. The process commonly known as "self-polishing" involves the application of a coating onto a surface base material, which is subjected to a continuous flow of seawater. This constant exposure to seawater results in the ionization of dissolved salts and a continuous alteration of the surface, leading to its self-polishing effect.

Upon its initial implementation, the coating quickly gained dominance in the antifouling coating industry [27]. Nevertheless, the persistent utilization of organotin antifouling coatings has revealed that tributyltin (TBT), despite its remarkable efficacy in preventing fouling, exhibits significant toxicity, particularly towards fish and shellfish. This toxicity poses a substantial threat to marine organisms, thereby endangering the marine ecosystem and potentially leading to species extinction [26]. In 2001, the International Maritime Organization (IMO) issued a declaration stating that the utilization of TBT antifouling coatings on commercial vessels worldwide would be banned, effective from 2008. Consequently, there has been a shift in research attention towards the advancement of environmentally friendly antifouling coatings. Subsequently, scholars employed organic copper, organic zinc, organic silicon, and various other elements [28–30] as alternatives to organotin in the formulation of antifouling coatings that are both tin-free and possess reduced toxicity. The grafts mentioned above exhibit a lower ionic antifouling capacity compared to organotin. Therefore, the inclusion of a copper antifouling agent is typically necessary during the preparation process to ensure a satisfactory antifouling effect [31].

The researchers have integrated acrylic acid and polyurethane copolymers that are capable of undergoing hydrolysis in seawater into the coatings, with the aim of developing environmentally sustainable self-polishing coatings that exhibit strong antifouling

characteristics [25]. Xuezhi Jiang and colleagues [32,33] from Wuhan University of Technology employed acrylic acid chloride as a chemical modifier for glyphosate, subsequently undergoing a sequence of polymerization reactions to produce polyacrylic acid resins featuring glyphosate side-linked branches. These resins were then subjected to fouling bio-inhibition experiments. The findings indicated that the polyacrylic acid resin, which featured glyphosate side-chained branches, exhibited an attachment inhibition rate of 41% for barnacle Venus larvae. Moreover, the highest observed attachment inhibition rate of 46.9% was achieved for Rhodophyta crescentica. The team led by Professor Xia Li at Ocean University of China [34] conducted a study where they introduced indole derivatives (NPI) as side chains into acrylic resins through the process of free radical polymerization. The researchers then evaluated the resulting resins for their self-polishing and antifouling properties. The findings of the study indicate that the copolymer exhibited significant antifouling characteristics due to the synergistic effect of the acrylate's self-polishing capability and the antifouling property of the indole derivative, as depicted in Figure 3B. Boron acrylate polymers were synthesized by Professor Rongrong Chen's team at the Harbin Institute of Technology [35]. Pyridine-diphenyl-borane with hydrolysis function was employed as the side chain in the synthesis process. The experimental findings indicated that these polymers exhibited enhanced antifouling properties against diatoms. Furthermore, the polymers demonstrated improved antifouling effects in suspension experiments conducted in the Yellow Sea of China. The research team led by Professor Chunfeng Ma from the South China University of Technology [36] synthesized a polyurethane with main chain degradability using *N*-2,4,6 trichlorophenylmaleimide (TCPM) through the integration of the mercapto-alkene reaction and condensation reaction. The subject of investigation involved conducting hydrolysis experiments to examine the release of TCPM in relation to the degradation of the polyurethane main chain. The experimental results demonstrated that TCPM was indeed released during this procedure. Simultaneously, the degradation rate of the material would exhibit a decrease as the quantity of TCPM content increases, thereby promoting the enhancement of antifouling material durability. The antifouling capability of a polyurethane material has been demonstrated through experiments conducted on marine pegboards. The observed phenomenon can be attributed to the regulated rate at which the material undergoes surface self-renewal, as well as the release of antifouling chemicals that occurs as a consequence of its degradation.

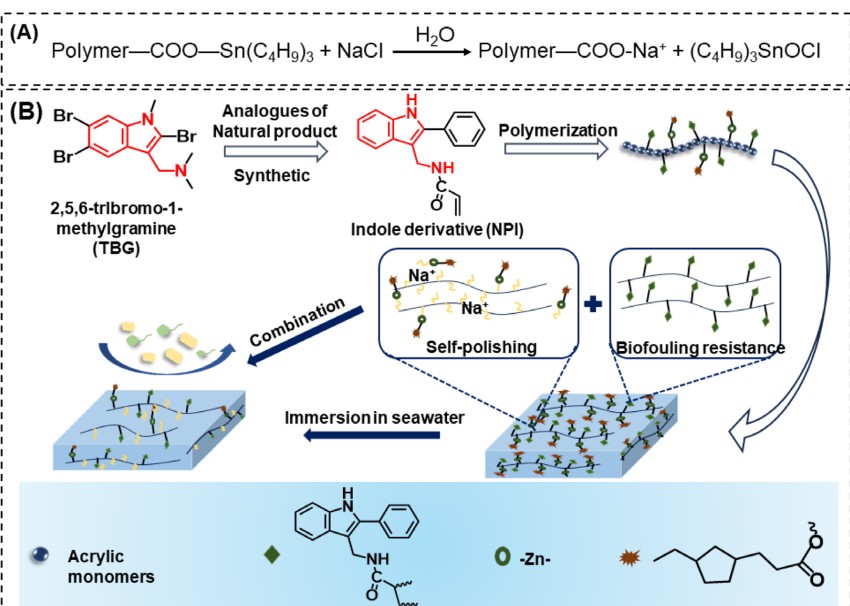

**Figure 3.** (**A**) Hydrolysis mechanism of organotin self-polishing resins; (**B**) Schematic diagram of the antifouling mechanism of NPI and acrylate resin copolymer [34] with permission (Copyright © 2023, Elsevier).

### 2.2. Fouling Release Antifouling Coatings

The efficacy of fouling-release antifouling coatings is primarily attributed to the inherent ability of marine fouling organisms to adhere to the surface of the coating and subsequently accumulate fouling materials with ease. Conventional fouling-release antifouling coatings possess low surface energy, thereby facilitating the removal of fouling through mechanical cleaning methods or minimal application of shear force. The aforementioned coatings exhibit consistent fouling-release properties and demonstrate enduring antifouling effects, as depicted in Figure 4A [37,38]. The efficacy of fouling-release antifouling coatings is typically influenced by various factors, including the thickness of the coating, its modulus of elasticity, surface roughness, and other parameters, as well as the material's hydrophobicity and low surface energy [39]. The antifouling performance of hydrophobic samples can be attributed to the inherent material properties and surface roughness, which can effectively trap air bubbles in the interstices. This, in turn, reduces the contact between protein organisms and the sample surface [40]. Fouling-release antifouling coatings have garnered significant attention in recent years due to their exceptional environmental compatibility [41,42]. In marine contexts, the implementation of this coating has the potential to mitigate the navigational resistance of ships by virtue of its smooth surface, thereby resulting in reduced economic consumption, such as fuel usage, and the enhanced longevity of ships, particularly with regard to propellers.

Currently, the two most commonly employed substances for fouling-release coatings are organ-silicone polymers and organ-fluorine polymers. This is primarily attributed to the limited strength of the interactions between chain segments containing silicone or fluorine and protein biomolecules, which renders them inadequate for the formation of biofouling [43,44]. In a study conducted by Yarbrough et al. [45] at the University of North Carolina at Chapel Hill, a set of perfluoro polymer brushes was fabricated on a glass substrate. The presence of functional groups, specifically perfluorooxymethylene, on the surface of the glass was found to significantly enhance its low-surface-energy properties. Consequently, the material exhibited improved resistance to protein adsorption and possessed fouling-release properties. Experimental investigations on the antifouling capabilities of the coatings demonstrated a remarkable 90% fouling release efficiency against microorganisms. The researchers led by Professor Chunfeng Ma from the South China University of Technology [46] developed a metal–ligand crosslinked organosilicon coating. The experimental results suggest that this coating exhibits enhanced self-healing and antifouling characteristics.

The aforementioned materials, which possess either hydrophilic or hydrophobic properties and have a single graft on their surface, still fail to achieve the intended antifouling effect. The development of a dirt-releasing coating has been achieved by researchers, which demonstrates the ability to prevent the attachment of fouling organisms and decrease the adhesion force between fouling and the surface of the material. This has been accomplished through the continuous optimization of the physical and chemical parameters of the coatings, as well as the rational design of the molecular structure. Furthermore, the coating exhibits effective protein adsorption prevention properties and possesses amphiphilic characteristics due to the presence of both hydrophilic and hydrophobic components. Additionally, it demonstrates certain capabilities for releasing dirt. The research team led by Professor Lingmin Yi at Zhejiang University of Technology [47] developed an amphiphilic polymer. This polymer is composed of a flexible fluorosilicon macromonomer, which serves as the hydrophobic component with a low surface energy, and an amphiphilic monomer, which acts as the hydrophilic component. This molecular structure is depicted in Figure 4B. The experimental results indicate that the coated surface effectively retained a substantial quantity of low-surface-energy fluoro silicone particles, even when immersed in water. A copolymer coating was synthesized by carefully adjusting the proportions of hydrophilic and hydrophobic surface segments, resulting in a coating with remarkable protein resistance. The researchers led by Junpeng Zhao from the South China University of Technology [48] synthesized amphiphilic polyurethane coatings with heterostructures

by incorporating poly(ethylene oxide) and birch alcohol. The experimental findings demonstrated that these coatings exhibited favorable characteristics in terms of protein adsorption resistance and fouling release properties. In a study conducted by Wooley et al. (University of Washington, USA), a polymer brush was synthesized using hydrophilic PEG chains and crosslinked hyperbranched fluorinated units. The researchers then performed simulated experiments using bovine serum protein as the fouling material [49]. The findings of the study indicated that the amphiphilic polymer brush exhibited enhanced efficacy in preventing protein adhesion, superior performance in resisting fouling, and improved ability to release fouling. In a work by Martinelli et al. (University of Pisa, Italy), they conducted a study in which they synthesized polystyrene polymer brushes containing amphiphilic side chains (PEG-b-PTFE). These brushes exhibited distinct phase regions attributed to the hydrophilic and hydrophobic components present. Upon exposure to water, the conformation of the brushes underwent changes, resulting in an increase in surface roughness and subsequent enhancement of inhomogeneity [50]. The results of the investigation into the degree of adhesion that various microorganisms and proteins showed to the surface suggested that the coated surface had a more significant antifouling effect.

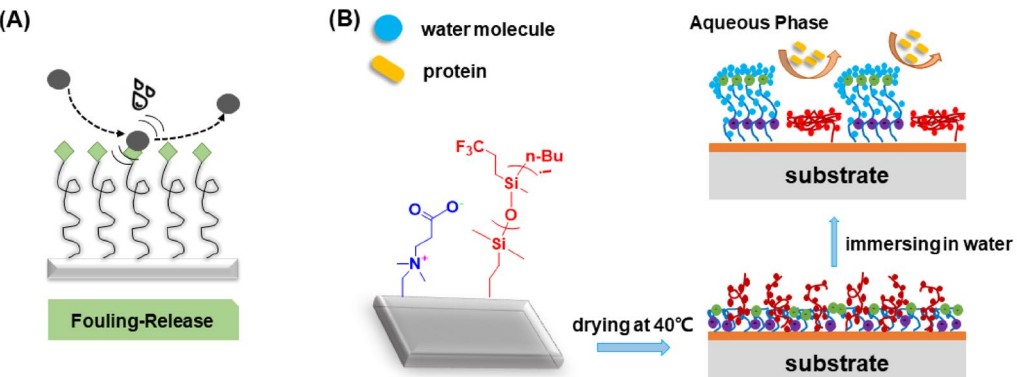

**Figure 4.** (**A**) Fouling release coating mechanism [51]; (**B**) Antifouling mechanism of amphiphilic copolymer [47] with permission (Copyright © 2023, American Chemical Society).

### 2.3. Bionic Antifouling Coatings

Numerous organisms in the natural environment possess remarkable antifouling characteristics owing to their distinct surface morphology and properties, rendering them highly resistant to the attachment of other organisms. As a result, when developing antifouling coatings, researchers frequently take inspiration from biological mechanisms. Presently, the prevailing body of research pertaining to biomimetic antifouling coatings can be categorized into two distinct groups. The initial approach involves the utilization of naturally derived antifouling agents, including terpene sugars emitted by certain marine organisms [52] and halofuranones [53], as well as extracts from terrestrial plants such as capsaicin [54] and carvacrol [55], all of which possess specific antifouling characteristics. One approach involves employing micro- and nano-construction techniques to replicate the biological surface properties of lotus leaves and sharks, as depicted in Figure 5A.

In the field of marine antifouling, there is an expectation that environmentally friendly antifouling agents will utilize natural compounds with comparable efficacy to effectively combat fouling. Nevertheless, there exist certain limitations in the present scenario, including intricate extraction processes and inadequate stability [55]. Consequently, it becomes imperative to establish a harmonious combination of natural antifouling agents and resin carriers to guarantee their stability and regulated release. The research team led by Jinggang Gai from Sichuan University [56] employed guanidine groups as antifouling agents in the fabrication of filter membranes. The results indicated that the membranes exhibited exceptional antifouling properties along with a notable selectivity in permeability. The research team led by Professor Peiyuan Qian from The Hong Kong University of Science and Technology [57,58] developed a butenolide-based antifouling agent. This agent was prepared by

utilizing polycaprolactone-based (PCL) polyurethane as a carrier, enabling a gradual and regulated release of the antifouling agent due to the biodegradable properties of the resin. The experimental results, depicted in Figure 5B, demonstrated that the developed agent exhibited a certain level of antifouling efficacy against larvae of marine fouling organisms. A research team led by Professor Xiaoli Zhan from Zhejiang University [59] employed a polycondensation reaction to graft modified coumarin and eugenol onto polyurethane side chains. This process was utilized to develop a polydimethylsiloxane (PDMS)-based marine antifouling polyurethane coating with smart properties. Based on the results, the layer exhibited enhanced mechanical properties, antibacterial properties, and resistance against algae growth. Additionally, it demonstrated a notable marine antifouling performance lasting for a duration of 9 months. These findings suggest promising prospects for the application of this layer in the field of marine antifouling.

The surfaces of shark skin and lotus leaves exhibit robust biofouling inhibition and self-cleaning properties, effectively preventing the attachment of fouling organisms such as barnacles and algae to the underlying substrate material. A collaborative effort between researchers from the University of Florida, USA, and the University of Birmingham, UK, resulted in the development of a microstructured antifouling coating inspired by shark skin [60]. This coating demonstrates a significant reduction of 85% in the attachment rate of macroalgae spores while also exhibiting a favorable effect on drag reduction. In their study, Zheng et al. conducted research at the Wuhan University of Technology to fabricate a polyurethane surface with lotus leaf characteristics [61]. This was achieved by utilizing the natural lotus leaf as a template through the replica molding method. The resulting surface exhibited a notable decrease in protein adsorption compared to its pre-construction state, resembling the lotus leaf's distinctive properties. In the realm of antifouling surface modification strategies, chemical surface modification is employed alongside physical alterations in the surface microstructure. The prevailing technique employed in current research is polymer brushing, characterized by a substantial polymer density, strong adherence to the water layer, and the ability to readily incorporate functional groups with anti-adhesion properties. The presence of the polymer creates a physical barrier that effectively maintains a predetermined distance between dirt particles and the surface of the material, thereby resulting in the attainment of antifouling properties. Qian Ye et al. [62] conducted a study at Northwestern Polytechnical University where they utilized 3D printing technology to create a surface resembling shark skin, which was then modified with a poly ionic liquid brush. The researchers used bovine serum protein as a model for fouling in their experiments, as depicted in Figure 5C. The findings indicated that the surface exhibited a superior anti-protein adhesion efficacy. The initial discovery of the remarkable anti-protein adhesion properties of polyethylene glycol (PEG) derivatives was made by Prime et al. [63] at Harvard University. The anti-protein adhesion properties of novel copolymers of poly (ethylene glycol) methacrylate (AEM-PEG) on glass substrates were discovered by Lonov et al. [64] from the Max Planck Institute in Germany. This finding is illustrated in Figure 5D. Additionally, the researchers highlighted the copolymers' advantageous features, including their low cost, ease of use, and compatibility with aqueous solutions. In a study conducted by Philip et al. [65] at the University of Texas, it was observed that the presence of bovine serum protein as a simulated fouling agent on the substrate surface was reduced to a minimum when the polyethylene glycol (PEG) content was 45%. This finding can be attributed to the decrease in the size of the split-phase region, which occurs as the PEG content decreases. The dimensions of an object have an impact on the ability of proteins and marine organisms, including microorganisms, to adhere to the material's surface.

Micro- and nano-structured antifouling materials predominantly leverage their microstructure to hinder the attachment of fouling organisms without causing any detrimental effects on the marine ecosystem. This contrasts with traditional antifouling techniques, which entail discharging antifouling substances. The majority of micro- and nanostructures are predominantly composed of soft materials, including silicon wafers, PDMS, and

similar substances. Currently, there is a scarcity of research pertaining to microstructured antifouling techniques specifically tailored for the hard metals employed in ship propellers. Moreover, the high method employed for such purposes is accompanied by certain limitations, including a complex procedural approach and substantial financial expenses. These factors collectively impede the widespread application of micro-nanostructures within the domain of ship propeller antifouling.

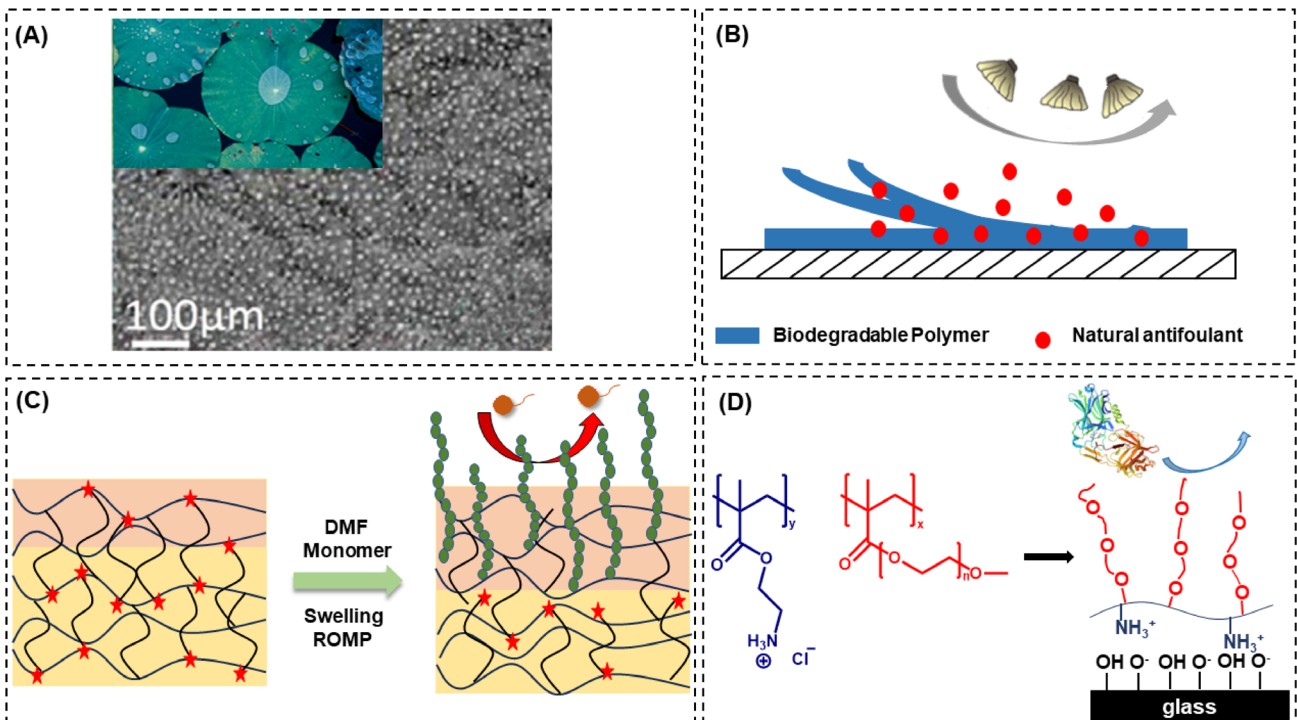

**Figure 5.** (**A**) Surface of lotus leaf and its SEM image [66] with permission (Copyright © 2023, American Chemical Society); (**B**) Mechanism diagram of natural antifouling agent to achieve antifouling performance using resin as a matrix [58] with permission (Copyright © 2023, American Chemical Society); (**C**) Mechanism of preparation of poly ionic liquid brush grafted imitation shark skin surface [62] with permission (Copyright © 2023,Elsevier); (**D**) Mechanism of anti-protein adhesion of linear PEG polymer brushes on glass substrates [64] with permission (Copyright © 2023, American Chemical Society).

## 3. Anti-Cavitation Coatings

The propeller, as the primary power system of a ship, exerts a substantial influence on the ship's operational lifespan owing to its prolonged and high-velocity operation, which gives rise to cavitation erosion. Currently, the mitigation of cavitation erosion in ship propellers is commonly addressed through two methods. Initially, the material undergoes surface modification treatment to enhance its anti-cavitation properties. Alternatively, research and development efforts are focused on exploring novel materials with superior characteristics in this regard. Nevertheless, the implementation of these modifications is limited due to the time-consuming and arduous nature of new material research and development. The second approach involves the application of material surface modification treatments, such as surface nitriding technology and surface coating technology [67]. These treatments have the potential to significantly mitigate cavitation damage in the overcurrent region by implementing anti-cavitation coatings. Surface coating technology has garnered significant attention due to its ability to provide long-term protection against cavitation-induced damage to materials [68]. Surface coating technology plays a significant role in mitigating cavitation effects. Based on the divergent characteristics of their constituent materials, the prevailing anti-cavitation coating technologies currently employed can be classified into two primary categories: organic coatings and inorganic

coatings, predominantly comprising metal coatings. The enhancement of the material's resistance to cavitation in organic coatings is primarily achieved through the utilization of the material's elastic properties to effectively absorb the energy generated during the cavitation process. Firstly, it is important to note that the application of metal coatings serves to enhance the fatigue resistance of the material surface, thereby improving its ability to withstand cavitation-induced damage. This is achieved through the utilization of the material's inherent hardness, relatively elevated mechanical properties, and resistance to high temperatures. The approach has been extensively developed.

*3.1. Cavitation-Resistant Organic Coatings*

In recent years, there has been a growing interest among domestic and international researchers in organic coatings for ship propellers and other overcurrent components that are required to function in seawater for prolonged durations. These coatings possess unique properties that effectively minimize the risk of corrosion in seawater. Moreover, they offer additional advantages such as affordability, ease of operation, and the ability to easily regulate their molecular structure. In recent years, there has been a significant increase in scholarly attention, both domestically and internationally, towards this subject matter. Currently, the most extensively studied polymer organic coatings for anti-cavitation purposes encompass polyurethane elastomers and polyurea elastomers. And considering the need for environmental protection, they use waterborne polymers more often. However, the organic coatings have poor abrasion resistance and low binding force with the surface of materials. So it is very vulnerable to falling off under the frequent impact of a periodic alternating cavitation force, which has become a major obstacle to application.

Polyurethane is a polymer material that consists of a diisocyanate-coupled polyether soft segment and a macromolecule diisocyanate hard segment. The soft segment imparts elasticity and flexibility to the system, while the hard segment allows for reversible crosslinking. Polyurethane exhibits favorable mechanical properties, such as resistance to abrasion and corrosion, along with the advantages of high strength and robust toughness [69]. Simultaneously, it is worth noting that polyurethane elastomer exhibits a notable loss factor, thereby facilitating the dissipation of a substantial amount of energy upon encountering external vibrations or impacts. This property effectively mitigates external harm and renders it a superior material for anti-cavitation purposes [70,71]. During the cavitation process, the molecular chain segments of polyurethane undergo softening due to the repetitive impacts from the airflow and microjet. Simultaneously, motion deformation takes place to absorb the energy generated by these impacts, as depicted in Figure 6A.

The initial application of polyurethane as a coating to prevent cavitation occurred in the year 1996. In a study conducted by Liangmin Yu et al. [72], a polyurethane coating was applied to propellers and other components to assess its effectiveness in mitigating cavitation. The findings indicated that the coating exhibited superior anti-cavitation properties and adhesion. However, it was observed that the coating had a shorter service life. Hydrophobic polydimethylsiloxane-based polyurethanes (Si-PUx) were synthesized by Professor Rongrong Chen's research team at Harbin Engineering University [73]. The synthesis involved a polycondensation reaction utilizing hydroxypropyl polydimethylsiloxane (H-PDMS) and polytetramethylene glycol (PTMG) as the soft segments and 2,4-toluene diisocyanate, 1,4-butanediol, and triethanolamine as the hard segments. Based on the results obtained, it was observed that the cavitation resistance of Si-PUx coatings exhibited a consistent increase with the progressive augmentation of H-PDMS content. However, a decrease in the adhesion of Si-PUs coatings was observed as the level of H-PDMS increased. The figure depicted in Figure 6B illustrates the surface morphology of Si-PUx and epoxy coatings subjected to varying cavitation durations. It is noteworthy that the Si-PUx coating, which incorporated 12.5 wt% of H-PDMS, exhibited a remarkably low accumulated mass loss of merely 2.96 mg. Following a duration of 80 h of cavitation testing, the surface demonstrated exceptional resistance to cavitation as it remained free from any observable fractures or voids throughout the entire testing period. In their study,

Qinghua Dai et al. [74] employed a three-layer composite coating approach to develop a cavitation-resistant polyurethane coating. The top layer consisted of polyurethane, while the intermediate layer was composed of a combination of epoxy and polyurethane interpenetrating resin. The primer layer, on the other hand, was made of epoxy resin. This composite coating exhibited enhanced mechanical properties and demonstrated superior adhesion to multiple substrates. The coating exhibited minimal surface degradation following 200 h of cavitation experiments, indicating superior resistance to cavitation. The cavitation resistance of polyurethane coatings reinforced with carbon nanofibers (CNFs) in seawater was examined by Lee et al. [75] through the application of the ultrasonic vibration method. The findings of the study demonstrated that the polyurethane coating lacking cellulose nanofibers (CNFs) exhibited the least resistance to cavitation. This suggests that the incorporation of CNFs has the potential to enhance the coating's ability to withstand the impact pressure resulting from the rupture of cavitation bubbles. Meanwhile, the polyurethane coatings that were supplemented with fluorine exhibited exceptional resistance to cavitation. The observed phenomenon of enhanced resistance to cavitation in polyurethane may be attributed to the synergistic effect between CNF and fluorine, which influences the material's structural properties. In a study conducted by Ning Qiu et al. [67] at Zhejiang University, various coatings including epoxy, ceramic, and polyurethane were applied onto rigid alloy surfaces in order to evaluate their resistance against cavitation. The primary factors influencing the cavitation durability of the coatings were found to be the adhesion and thickness of the coatings. Furthermore, upon analyzing the degradation mechanism of the coatings, it was determined that polyurethane coatings exhibited a prolonged latent period, thereby enhancing the materials' resistance to cavitation.

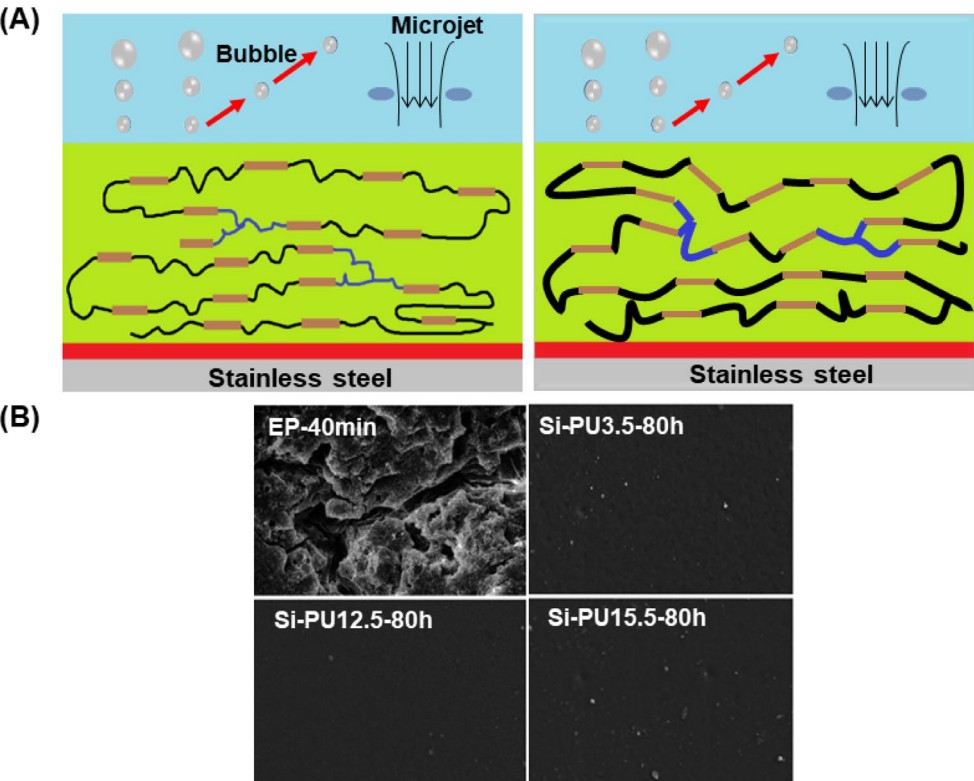

**Figure 6.** (**A**) Cavitation resistance mechanism of polyurethane elastomer coating [73]; (**B**) SEM image of Si-PUx and epoxy coatings at different cavitation times [73] with permission (Copyright © 2023, Wiley).

The emergence of polyurea as a corrosion-reducing coating with impact-resistant properties was initially observed in the 1980s [76]. Polyurea is a type of elastomeric substance that is produced through the chemical reaction between isocyanate groups and

amino compounds. This material is characterized by its solvent-free nature, making it environmentally friendly. Additionally, it exhibits remarkable resistance to moisture and humidity, along with exceptional mechanical properties, including good corrosion and weather resistance. Furthermore, this material finds widespread application in various domains such as corrosion resistance, abrasion resistance, and waterproofing [76,77]. In their study, Shi Feng et al. [78] synthesized a range of polyurea anti-cavitation coatings by varying the concentration of MOCA, HDI trimer prepolymer, and polyaspartic ester. MOCA was employed as the chain extender in this two-component system. The findings indicate that the polyurea coatings produced exhibit favorable flexibility and notably superior resistance to cavitation compared to metal coatings. However, it is worth noting that these polyurea coatings have a relatively restricted lifespan. The researchers Marlin et al. [79] discovered that the failure of the coating was a result of the combined impact of locally transmitted impulse load and material heating. This conclusion was drawn based on their examination of the cavitation surfaces of the polyurea coatings. Furthermore, the resistance of the coating to cavitation is influenced by various factors, including the polyurea composition, the intensity of the cavitation field, the substrate material, the coating thickness, and several other contributing elements. Corrosion and temperature are also affected by these factors.

In comparison to polyurethanes, polymers featuring urea bonding exhibit enhanced responsiveness, thereby expediting the spraying procedure for polyurea coatings and simplifying their application on large-scale equipment such as ships. In the actual cavitation process, the use of polymer coatings may have some small polymer particles shedding into the ocean, but waterborne polymers do not have a significant impact on the marine environment and organisms, and most of them are environmentally friendly.

### 3.2. Cavitation-Resistant Inorganic Coatings

Inorganic coatings designed to mitigate cavitation on ship propellers encompass various options, such as copper alloy coatings, stainless steel coatings, nickel plating, and ceramic coatings. The predominant techniques employed in the modification of propeller surface metal anti-cavitation coatings consist of thermal spraying technology, surface laser modification technology, surface nitriding technology, and surface plasma modification technology, among others.

The phrase "thermal spraying technology" encompasses the procedure of elevating the temperature of the material intended for spraying until it reaches a molten or partially molten state. Subsequently, this material is expelled at a predetermined velocity to form a protective layer on a previously prepared substrate. The process exhibits traits such as operational flexibility, a high rate of material deposition, and the ability to control the thickness of the coating. Currently, the thermal spraying materials that are predominantly utilized encompass cobalt-based alloys, nickel-based alloys, and other similar materials [80]. In their study, Gu et al. [81] employed Cu95 as the coating material and employed thermal spraying technology to conduct propeller corrosion experiments. The findings of the study indicate that the coatings mentioned above exhibit exceptional resistance to cavitation and corrosion in seawater. Consequently, these coatings can be effectively employed as protective surface coatings for propellers made of manganese-copper alloy. Amarendra and colleagues [82] fabricated a thermal spray coating consisting of 70% nickel and 30% chromium through the utilization of the high-velocity oxygen fuel (HVOF) process. The researchers subsequently conducted an examination and comparison of the cavitation resistance between the thermal spray coating and the uncoated martensitic stainless steel (ss410). Furthermore, the researchers conducted experiments to assess the hardness, bending, and peeling characteristics of the specimens. The results indicated that, when subjected to identical conditions, the coated specimens exhibited significantly higher resistance to cavitation compared to the uncoated specimens. Moreover, the cavitation resistance of stainless steel 410 (ss410) can be significantly enhanced through the application of high-velocity oxygen fuel (HVOF) coating, which employs a combination of brittle and ductile erosion mechanisms. The afore-

mentioned findings were revealed as a result of the analysis conducted on the specimens. In their study, Szala et al. [83] employed supersonic flame spraying as a technique to apply HVOF coatings of MCrAlY and NiCrMoFeCo onto a stainless steel substrate, specifically AISI310 (X15CrNi25-20). Subsequently, cavitation experiments were conducted using the vibration method. The results of the study indicate that the MCrAlY coating exhibited a comparatively reduced level of wear resistance in comparison to the NiCrMo coating. Meanwhile, the sliding wear resistance of the alloy demonstrates enhancement as the nickel content proportionately increases.

The technology of surface laser modification has its origins in the 1960s. It involves the utilization of high-energy density laser irradiation to treat the surface of metallic materials, resulting in the formation of an alloy layer with distinct properties. The introduction of alloy layers has the potential to alter the structural arrangement, hardness, density, and uniformity of the material, thereby enhancing its resistance to cavitation. Furthermore, the preservation of the material's bulkiness facilitates the development of the alloy system's microstructure and the formation of substable phases. The primary techniques encompassed within this method are laser surface alloying (LSA), laser cladding (LSC), laser melting condensation (LSM), and various other processes [84]. In their study, Lian Fen et al. [85] employed surface laser modification as a technique to create texturization on Ti6Al4V alloy. The grid-textured specimens exhibited superior strength and cavitation resistance compared to the straight-textured specimens, owing to their elevated surface hardness, broader hardness gradient, and even distribution of prominent high-hardness regions.

Laser surface alloying (LSA) is a technique that involves the use of laser light and solid-phase material to perform directional energy-beam-assisted surface alloying. The thermal phenomenon resulting from the interaction between the laser and solid-phase substances leads to the melting of the metal surface, including the alloying elements that have been introduced. Subsequently, rapid condensation occurs, resulting in the formation of a coating on the substrate's surface. The alloying elements can be incorporated using either the direct injection method or the pre-painted coating method [86]. In the study conducted by Yi [87], laser alloying technology was employed to fabricate high entropy alloyed coatings with diverse compositions on the surface of 304 stainless steel while ensuring controlled atmosphere conditions. Following a 5 h cavitation process, it was observed that the high entropy alloyed coating exhibited no material spalling on the surface of the sample material. Additionally, there was no apparent plastic deformation and a significant reduction in the cavitation phenomenon. These findings indicate that the coating demonstrates superior resistance to cavitation when compared to 304 stainless steel. Laser cladding (LSC) refers to the process of applying an alloy or composite material as a protective layer onto the surface of a substrate. The production of protective surface materials can greatly benefit from its extensive range of potential applications and value. To improve the mechanical properties and resistance to cavitation of 17-4PH stainless steel, Ding et al. [88] employed laser technology to implement two different surface treatments. One method entails the laser-induced solidification of tungsten-chromium-cobalt alloy powder, resulting in its hardening, whereas the other method involves the application of a laser coating onto the alloy surface. The findings indicate that the application of heat treatment and laser cladding techniques can potentially enhance the hardness, Young's modulus, resistance to plastic deformation, and resistance to cavitation of the alloy. Although the laser cladding treatment did not significantly enhance the hardness of the steel, it outperformed the laser heat treatment in terms of augmenting the steel's Young's modulus and resistance to cavitation. In a study conducted by Zhang Song et al. [89], a laser melting technique was employed to apply a NiCrBSi coating onto the surface of an aluminum alloy. The objective of this approach was to enhance the material's resistance against cavitation. The application of the laser melting solidification (LSM) technique results in an improved resistance to cavitation by promoting a more uniform distribution of applied force. Ren Yuhang [90] employed a combination of laser cladding technology and laser melting coagulation technology. Initially, a laser melting coagulation treatment was performed on the surface of stainless

steel, followed by a subsequent treatment using laser melting cladding technology on NiTi alloy. This approach resulted in an enhancement of the material's resistance to cavitation.

The term "surface nitriding technology" encompasses various methods such as plasma nitriding and ion nitriding, which are employed to modify the surface properties of a material. These techniques aim to introduce a durable nitride coating that can effectively integrate with the substrate, thereby improving the material's resistance to cavitation. In a study conducted by Huang et al. [91] from Feng Chia University in Taiwan, it was observed that ion nitriding had a significant effect on enhancing the cavitation resistance of carbon steel. The cavitation resistance of 316L stainless steel in seawater was examined by Chong et al. [92] following ion nitriding at various temperatures. The plasma nitriding process was conducted with a $N_2$ to $H_2$ ratio of 1:4 at temperatures ranging from 400 to 500 °C for a duration of 10 h. An improvement in cavitation resistance was observed as the nitriding temperature was raised from 400 °C to 500 °C. When comparing the untreated samples to the treated ones, it was observed that the material exhibited a notable reduction in weight loss and damage rate within the temperature range of 400 to 500 °C. Additionally, the hardness, mechanical properties, and resistance to cavitation of the material exhibited a significant increase. In a study conducted by Szkodo et al. [93], stainless steel was subjected to nitriding treatment. The researchers observed that the surface layer of the nitrided stainless steel exhibited significantly enhanced cavitation resistance compared to the reference samples. In their study, Mitelea et al. [94] employed gas nitriding technology to fabricate coatings on the surface of aluminum alloys. This approach demonstrated a significant enhancement in the cavitation resistance of the alloys.

Currently, the predominant technique for surface plasma modification involves the introduction of metal ions, such as nitrogen (N) or boron (B), into the plasma to enhance surface hardness. Another commonly employed method is cathodic arc ion plating, which results in the deposition of a wear-resistant plating layer with high hardness. These modifications serve to enhance the material's resistance to cavitation. The cavitation resistance of cathodic arc ion plating is significantly influenced by the strength of the bond. Based on the findings of Krella et al. [95], it was observed that the TiN coating exhibits optimal bonding strength with the substrate at a temperature of 350 °C. Conversely, the CrN layer demonstrates superior bonding strength with the substrate at a temperature of 500 °C, resulting in reduced mass loss and improved resistance against cavitation.

In brief, the application of inorganic coatings has demonstrated the potential to mitigate material and economic detriments arising from cavitation, thereby enhancing anti-cavitation efficacy to a certain degree. However, in real-world scenarios encompassing corrosive settings like the ocean, as well as in critical components such as propellers necessitating prolonged operation at high velocities amidst overcurrent circumstances, metal coatings are susceptible to electrochemical corrosion, thereby diminishing their longevity to a certain extent. Simultaneously, the process requirements for these surface treatment technologies are comparatively stringent, thereby rendering their implementation more challenging. When comparing metal coatings to organic coatings, it is observed that organic coatings possess several advantages. These advantages include lower cost, increased operational flexibility, and the potential for certain organic coatings to exhibit elasticity. This elasticity enables the absorption of energy released during the cavitation process, thereby enhancing the material's resistance to cavitation. The issue of cavitation resistance has garnered significant interest in the realm of ship propeller protection in recent times. The utilization of organic coatings for this objective has witnessed a growing trend.

## 4. Integration of Antifouling and Anti-Cavitation Coatings

In the context of surfaces such as ship overflow parts and hydraulic turbine propeller blades, it is commonly observed that biofouling and cavitation corrosion occur concurrently, with an evident interplay between these phenomena. The presence of fouling organisms on a material leads to a decrease in its performance, as it promotes the formation of surface vacuoles. This, in turn, exacerbates the occurrence of cavitation-induced damage to the

material. Additionally, the chloride ions in seawater can penetrate and destroy the surface of the materials and make them prone to localized corrosion such as pitting corrosion. The occurrence of cavitation erosion exacerbates the surface roughness of materials, such as propellers, thereby promoting the adhesion of marine fouling organisms. This creates a self-perpetuating cycle of fouling attachment [96].

In relation to the present market conditions, there is an expectation that the demand for antifouling coatings will exhibit a growth rate of 10% from 2021 to 2025, resulting in a market value of USD 1.96 billion. Notably, the Asia–Pacific region is projected to experience a substantial increase of 51%, equivalent to an 8.66% growth in 2021 alone. The demand for anti-cavitation coatings in the Asia–Pacific region accounts for 33% of the overall market demand. It is projected that the global demand for these coatings will experience a growth rate of 5% in the coming decade. By the year 2027, it is anticipated that the global market demand for anti-cavitation coatings will exceed USD 20 billion. Hence, the global imperative for the development of antifouling and anti-cavitation coatings is evident.

In 1987, Japanese scientists conducted an experiment involving the application of self-polishing antifouling coatings and silicone synthetic resins to propeller blades. Following 500 h of operation, the propeller blades exhibited minimal adherence of seafood and peeling of the paint film, thereby maintaining a notably clean condition [97]. The silicone coatings Intersleek ® 700 and fluoropolymer coatings Intersleek ® 900 were developed by the Netherlands International Paint Company (IP) as antifouling coatings with low surface energy properties. The application of a series of coatings was initially implemented on submarine propellers in 1995. Following a period of 12 months of utilization, it was observed that the coating remained in an exceptional condition. The Belzona2141 polyurethane resin coatings, manufactured by the British Belzona Company, are designed for application on various components such as propellers, turbines, valves, and other overflow equipment. The construction and operation of these devices are characterized by simplicity while also exhibiting a commendable degree of cavitation and corrosion resistance. The Metaline® series 700 two-component polyurethane coatings were manufactured by Germany's MetaLine Company. The findings indicated that the product exhibited improved resistance against fouling and cavitation simultaneously. Currently, there has been limited advancement in the incorporation of foreign antifouling and anti-cavitation coatings. This progress has primarily been observed in military applications, which do not encompass a significant portion of commercial and civil usage. Conversely, research on domestic antifouling and anti-cavitation integrated coatings is still in its early stages.

In the year 2011, scientists from China submitted a patent application with the objective of mitigating the corrosion of ship propellers and the accumulation of marine organisms. The initial step involved the removal of the oxide skin from the surface of copper alloy propellers to achieve a textured surface. Subsequently, a ceramic insulating coating composed of metal oxides was prepared using the thermal spraying technique. Following this, a metal antifouling coating was thermally sprayed onto the prepared ceramic coating, resulting in a composite coating that possesses both fouling and anti-cavitation characteristics. In their study, Weiwei Cong et al. [98] conducted research at the State Key Laboratory of Marine Coatings to develop a fouling release protective coating for propellers. The coating was prepared using an anticorrosive primer, elastic buffer paint, intermediate connecting paint, and antifouling topcoat. The researchers asserted that this coating exhibited both safety and environmental friendliness for construction personnel and the marine environment. Furthermore, they reported that the coating demonstrated exceptional efficacy in inhibiting the adhesion of marine fouling organisms and exhibited strong anti-cavitation performance. The assessment of the static antifouling efficacy was conducted at the Qingdao Zhongkang offshore test station, revealing that the surface condition remained favorable even after a duration of 36 months. In their study, Haocheng Yang et al. [99] conducted the synthesis of polyurethane (PU)/ZIF-8 (PHZ) composite coatings using a nanocomposite approach. The researchers utilized a multi-scale zeolite imidazolium skeleton material (ZIF-8) as a nanofiller, which is known for its environmentally friendly properties. The findings indicate that

the ZIF-8 polyurethane coating exhibited superior antifouling and anti-cavitation properties at a particle size of 50 nm. The coverage of the selected fouling agent, small crescent-shaped rhododendron algae (Nitzschia clostridium), in the simulation experiment was found to be only 0.51%, as depicted in Figure 7A. The mass loss of the coating during the 30 h cavitation experiment amounted to a mere 9.9 milligrams. The incorporation of nanoparticles resulted in the improvement of the coating's hydrophobic properties, thermal stability, and mechanical characteristics. Nevertheless, it should be noted that ZIF-8 exhibits a diminished level of durability when exposed to saltwater, necessitating a gradual release of zinc ions over an extended duration in order to attain optimal antifouling efficacy. This, in turn, increases the probability of potential contamination in the secondary marine environment. Poly (dimethylsiloxane etherimide) (APT-PDMS) and poly (tetrahydrofuranediol) (PTMG) were utilized as raw materials in their study [100]. To introduce perfluorohexanediol (PFHT) into the polyurethane backbone, a two-step polymerization reaction was employed to synthesize a range of fluorinated isocyanate prepolymers (FIPs) with varying contents. This process resulted in the production of a fluoro silicone-containing polyurethane elastomer coating known as SFPU-x. The incorporation of fluorinated isocyanate prepolymers (FIPs) facilitates the formation of fluorinated polyurethane microdomains, thereby enabling the generation of microstructures with low surface energy. This is achieved through the enhancement of hard-segmented microzone structures within the coating. Based on the findings, it was observed that the SFPU-5 coating containing 5% FIP exhibited the most optimal level of internal microphase separation. The surface of this coating displayed a distinct microstructure, a significantly high water contact angle, and evident hydrophobic properties, as depicted in Figure 7B. These characteristics effectively hindered the adhesion of proteins and algae. The fouling simulation using bovine serum proteins resulted in a coverage of only 1.3% after a duration of 72 h. Additionally, the surface showed no discernible holes or cracks after 10 h of cavitation. The cumulative mass loss in deionized water and seawater was measured to be 2.7 mg and 2.9 mg, respectively, indicating a high level of cavitation resistance. Additionally, the investigation revealed that the microstructural characteristics of the coating surface have a significant impact on its ability to resist fouling and cavitation. The prevention of microorganism reproduction on the coating surface can be attributed to the presence of an inhomogeneous microstructure. This microstructure hinders the microorganisms' ability to reproduce through the "size-matching" effect and enrichment, thereby impeding their attachment and accumulation on the surface. The bubbles generated during the cavitation process undergo simultaneous collapse upon reaching the surface of the irregular microstructure. The region of the microstructure that is situated at its uppermost portion frequently experiences the lowest local pressure. Consequently, the area characterized by low external pressure is where the high internal pressure cavitation will ultimately collapse. The implosion force will be absorbed by the coating and transformed into both elastic and plastic deformation. This transformation leads to a reduction in the impact force, thereby preventing the occurrence of cracks and enhancing the quality of the loss, as depicted in Figure 7C.

For the sake of concision, it is clear that there is a sizable market need and substantial research relevance for the creation of an integrated coating that offers ship propellers both antifouling and anti-cavitation capabilities. However, the existing body of research on this topic is limited, necessitating a substantial number of studies to comprehensively understand and optimize the performance of propellers with the desired characteristics of "dynamic anti-cavitation and static fouling resistance". In light of the growing emphasis on environmental preservation, there has been a heightened interest in utilizing eco-friendly substances for various purposes. For instance, numerous antifouling agents found in nature, such as capsaicin, carvacrol, and halogenated furanone, have been identified as potential alternatives. Additionally, biomimetic materials, such as lotus leaves, sharkskin, and dolphins, have also garnered attention for their potential applications in this domain. They can be employed to fabricate high-quality integrated coatings that possess both antifouling and anti-cavitation properties. Furthermore, it is feasible to fabricate coatings that possess

anti-cavitation properties and are functionalized with antifouling agents. This can be achieved through the process of gradient compounding, wherein antifouling functionalized nanoparticles are incorporated into the matrix. This approach can be explored from two research perspectives: modification of micro and nano functionalized fillers, as well as the design of the polymer chain segment structure.

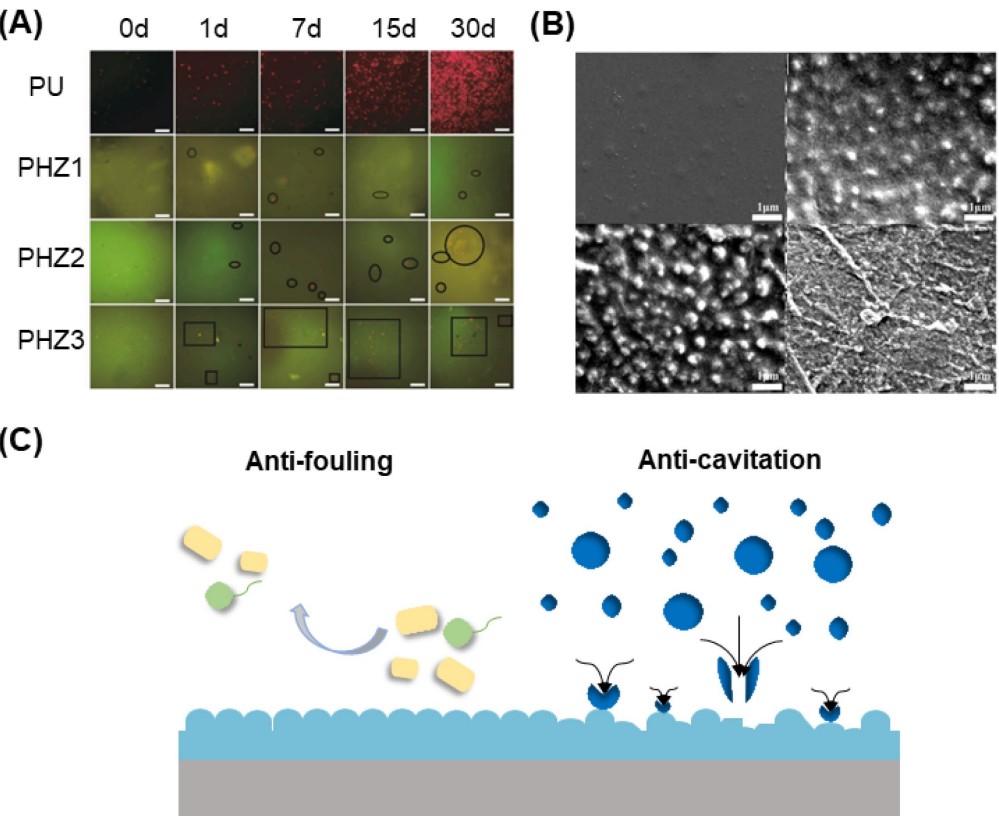

**Figure 7.** (**A**) Fluorescence microscope images of algae on PU, PHZ1, PHZ2, and PHZ3 coatings at different immersion times with a scale of 50 μm [99] with permission (Copyright © 2023, Elsevier); (**B**) SEM image of the microstructure of polyurethane-coated SFPU-x surface with a scale bar of 1 μm [100] with permission (Copyright © 2023, Elsevier); (**C**) Mechanism of antifouling and anti-cavitation of microphase separation structure.

## 5. Conclusion and Prospects

In brief, the global research community is actively engaged in investigating and addressing the challenges associated with ship propeller technology, with a particular focus on dynamic anti-cavitation and static fouling. The condition of the propeller has a direct influence on the ship's performance and lifespan, particularly as the maritime environment undergoes changes and the ship's service life extends. There is currently a disparity between domestic and international antifouling and anti-cavitation integrated technologies, hindering the full understanding and implementation of environmentally friendly coatings in this field. There exist numerous unresolved matters that necessitate the attention and ongoing investigation of scientific and technological professionals. These concerns primarily manifest in the subsequent domains.

Firstly, previous research has focused on addressing various issues related to the antifouling coating, such as enhancing its efficiency through functionalization, ensuring its environmental friendliness, and improving its strength and adhesion to counteract cavitation effects. Previous studies have proposed various solutions for addressing the issue at hand. These include the utilization of fouling release antifouling coatings and bionic antifouling coatings as alternatives to the environmentally harmful self-polishing antifouling coatings, with the aim of achieving enhanced antifouling efficacy. Simulta-

neously, the utilization of organic materials instead of inorganic ones can contribute to the mitigation of corrosion, particularly in seawater environments. This approach offers advantages over inorganic coatings, which are more susceptible to the flaws associated with electrochemical corrosion. This will potentially enhance the durability of the coating to a certain degree.

Furthermore, it is imperative for future researchers to prioritize the exploration of practical strategies aimed at comprehensively resolving the diverse challenges associated with integrated coatings for antifouling and anti-cavitation purposes. By employing rational molecular structure design and surface modification techniques, as well as incorporating functional fillers through compounding, it is possible to enhance the antifouling and anti-cavitation properties of ship propellers. Simultaneously, this approach can also strengthen the bond between the integrated coating and the underlying base material. This enables us to enhance and evaluate ship propeller protective coatings that possess the capability to deliver exceptional performance and extended durability in real-world marine conditions.

**Author Contributions:** Conceptualization, J.Z. and W.Q.; software, D.W. and Z.F.; investigation, W.C.; resources, L.L.; writing—original draft preparation, J.Z.; writing—review and editing, W.C. and J.Z.; visualization, J.Z.; supervision, Y.W.; project administration, W.Q. All authors have read and agreed to the published version of the manuscript.

**Funding:** This research received no external funding.

**Institutional Review Board Statement:** Not applicable.

**Informed Consent Statement:** Not applicable.

**Data Availability Statement:** Available in the paper.

**Conflicts of Interest:** The authors declare no conflict of interest.

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
