# Peer review of "Integration of Antifouling and Anti-Cavitation Coatings on Propellers: A Review"

_coatings, doi:10.3390/coatings13091619_

Round 1

Reviewer 1 Report

This contribution deals with a review on coatings to protect marine parts like propellers from fouling and cavitation erosion. Some weaknesses are listed below.

1.    As a matter of fact, the style of review papers sometimes resembles storytelling rather than a scientific text. This is intensified by use of colloquialism. As a consequence, it is suggested to revise the contribution in terms of language with the help of a native speaker.

2.    When writing a review about material degradation by corrosive or mechanical stress, the authors should be very precise in the application of appropriate vocabulary as well as definition of different terms. Some examples:

a.    “Void” and “scour” corrosion seem to be quite uncommon expressions. Should it rather be “localized” or “pitting corrosion”?

b.    Cavitation occurring near a material surface inevitably results in mechanical (!) damage, the corresponding type of wear is called “cavitation erosion”. It is common knowledge that cavitation erosion is caused by mechanical stress. However, it can be accompanied and influenced by corrosive attack of the liquid, which i.e. can contribute to the overall material degradation by hindering the self-healing of a passive layer in the case of a stainless steel. If so, the resulting type of degradation is called “cavitation corrosion” or “cavitation erosion-corrosion”.

c.     As a consequence of the last point, the pits induced by cavitation (due to the action of shock waves and / or microjets) must not be named an etch pit.

3.    Figure 1: from these sketches, the “mechanisms” of “anti-fouling” and “anti-cavitation” (which should rather be “anti-cavitation erosion”) cannot be understood.

4.    “The phenomenon of cavitation was initially observed and postulated in 1987 by the 101 esteemed British scientist S.W. Barneby during an investigation into the propeller effi-102 ciency of maritime vessels [20].” This is most probably wrong. It should have been around 100 years earlier.

5.    The use of polymer coatings, which are likely to be removed by cavitation erosion, will probably result in entry of small polymer particles into the ocean. Is there any focus or knowledge on environmental impact of such coatings?

6.    To sum up, the entire contribution requires a revision with careful focus on high preciseness in terms of language, use of correct terms and definitions, and accurate literature review.

Please see comment no. 1.

Author Response

Response to Reviewer 1 Comments

Point 1: As a matter of fact, the style of review papers sometimes resembles storytelling rather than a scientific text. This is intensified by use of colloquialism. As a consequence, it is suggested to revise the contribution in terms of language with the help of a native speaker.

Response 1: We are appreciated for your suggestions. We have tried our best to improve the language in the revised manuscript. These changes will not influence the content and framework of the paper.

Point 2.a: “Void” and “scour” corrosion seem to be quite uncommon expressions. Should it rather be “localized” or “pitting corrosion”?

Response 2.a: We are really sorry for our carelessness. We have explained the change made, including the exact location where the change can be found in the revised manuscript.

Localized corrosion is characterized by damage that occurs preferentially at discrete sites on a material surface and may result in the formation of pits, cracks and grooves. Pitting is a form of localized corrosion damage that results in the formation of small defects or pits, often associated with a cover of corrosion product or perforated metal over the pit [1].

The marine environment contains more corrosive factors than other naturally occurring conditions. It is usually more aggressive than inland environments due to the high concentration of chloride ions in seawater, which can penetrate and destroy passive protective films on metal alloys and make them prone to localized corrosion such as pitting corrosion [2].

Point 2.b: Cavitation occurring near a material surface inevitably results in mechanical (!) damage, the corresponding type of wear is called “cavitation erosion”. It is common knowledge that cavitation erosion is caused by mechanical stress. However, it can be accompanied and influenced by corrosive attack of the liquid, which i.e. can contribute to the overall material degradation by hindering the self-healing of a passive layer in the case of a stainless steel. If so, the resulting type of degradation is called “cavitation corrosion” or “cavitation erosion-corrosion”.

Response 2.b: We feel sorry for our mistakes. In our resubmitted manuscript, the mistakes are revised. And we also added some information below.

Cavitation is a unique form of erosion-corrosion that almost every propeller suffers from. It occurs at high velocities and fluid dynamic conditions, where significant pressure variations exist [3].

Erosion is a type of mechanical-related corrosion. It damages the passive films on the metal surface due to the high relative movement between the corrosive media and the metal. The corrosion rate increases due to the constant removal of the passive films and the exposure of the fresh metal surface. Therefore, the rate of erosion significantly depends on the velocity of seawater. Usually, this type of corrosion can be characterised by a special grooved appearance and a directional surface pattern. Corrosion wear is a type of erosion that occurs when there are solid particles in the corrosive media. Thus, corrosion wear is a combined effect of electrochemical corrosion and mechanical wear that significantly deteriorates the surface and increases the damage to the material compared with electrochemical corrosion alone.

Propeller cavitation in the marine environment involves the simultaneous existence of mechanical-related and electrochemical corrosion, which interact synergistically to accelerate the corrosion of one another (i.e., cavitation erosion-corrosion) [2].

Moreover, when the cavitating fluid is corrosive, the material loss is not purely mechanical in nature because corrosion also comes into play. When cavitation occurs in corrosive media, erosion-induced corrosion and/or corrosion-induced erosion will intensify the damage process and termed as ‘cavitation erosion–corrosion’. Erosion and corrosion often occur synergistically and material loss can be markedly higher than the sum of the effects of the processes acting separately [4].

Point 2.c: As a consequence of the last point, the pits induced by cavitation (due to the action of shock waves and / or microjets) must not be named an etch pit.

Response 2.c: We feel sorry for our mistakes. Cavitation means the formation of bubbles or cavities in a liquid due to reduction in local pressure in the liquid. It is the consequence of Bernoulli's equation, which states that when the flowing speed of a liquid increases, its pressure decreases. When the local pressure drops below a critical value, bubbles will form. When these bubbles encounter a high local pressure, they will implode, generating micro-jects or shock waves. When the implosion of bubbles occurs near a solid surface, these micro-jects or shock waves impart intense pressure to the solid surface. Upon repetition of such events, the surface region under attack will undergo fatigue and rupture, with material loss from the surface. This is known as cavitation erosion (CE). CE is thus caused by the localized cyclic impact of fluid against a surface during the collapse of cavities. In metallic materials accumulated work-hardening and crack formation are commonly observed. In some cases when the cavitation is intense, the density of cavitation pits is high enough to make a porous matrix and finally destroyed the component [4].

Point 3: Figure 1: from these sketches, the “mechanisms” of “anti-fouling” and “anti-cavitation” (which should rather be “anti-cavitation erosion”) cannot be understood.

Response 3: We think this is an excellent suggestion. We have redrawn Figure 1 according to your comment, as shown below.

Point 4: “The phenomenon of cavitation was initially observed and postulated in 1987 by the 101 esteemed British scientist S.W. Barneby during an investigation into the propeller efficiency of maritime vessels [20].” This is most probably wrong. It should have been around 100 years earlier.

Response 4: We are really sorry for our careless mistakes. Thank you for your reminder. The phenomenon of cavitation was initially observed and postulated in 1887 by the esteemed British scientist S.W. Barnaby during an investigation into the propeller efficiency of maritime vessels. And also the reference has been changed to: Barnaby, S.W. Marine Propellers; Being a Course of Three Lectures delivered at the Royal Naval College.; E. & F.N. Spon: London, British, 1887.

Point 5: The use of polymer coatings, which are likely to be removed by cavitation erosion, will probably result in entry of small polymer particles into the ocean. Is there any focus or knowledge on environmental impact of such coatings?

Response 5: We have added this part in the revised manuscript and supplemented extra data.

These small plastic bits are called “microplastics”. Microplastics, as the name implies, are tiny plastic particles. Officially, they are defined as plastics less than five millimeters (0.2 inches) in diameter smaller in diameter than the standard pearl used in jewelry. There are two categories of microplastics: primary and secondary. Over the past few years, scientists have begun to realize that the increasing volume of plastic materials slowly decomposing in the world’s oceans may present a long-term problem for marine food chains already reeling from over fishing and other anthropogenic insults [5].

The commonly used plastics include polystyrene, nylon, polyurethane, polypropylene, etc. these plastics are gradually decompose by the physical, chemical and biological effects in the environment, plastics are easily fragmented under the effect of environmental forces, however, it takes about a long time for these plastics to be completely decomposed. Most plastics will form plastic debris with a small particle size. The microplastics pollution has caused many hazards to marine life, and has already aroused widespread concern [6]. The following chart shows the toxicological effects of microplastics on fish. In this table, we are not aware of any hazards of organic anti-cavitation coatings matericals, such as PU.

Table 1. Toxicological effects of microplastics on fish.

And in the actual cavitation process, the organic coatings have poor abrasion resistance and low binding force with the surface of materials. So it’s very vulnerable to fall off under the frequent impact of periodic alternating cavitation force, which has become a major obstacle to the application [7]. The use of polymer coatings may have some small polymer particles shedding into the ocean, but waterborne polymers do not have a significant impact on the marine environment and organisms. So considering the need of environmental protection, they use waterborne polymers more often.

Point 6: To sum up, the entire contribution requires a revision with careful focus on high preciseness in terms of language, use of correct terms and definitions, and accurate literature review.

Response 6: According to your comments, we have made extensive modifications to our manuscript and supplemented extra explanation to make our results convincing. Thank you again for the valuable comments that we have used to improve the quality of our manuscript.

Reference:

[1]          LYON S. 1 - Overview of corrosion engineering, science and technology [M]//FéRON D. Nuclear Corrosion Science and Engineering. Woodhead Publishing. 2012: 3-30.

[2]          WANG A, DE SILVA K, JONES M, et al. Anticorrosive coating systems for marine propellers [J]. Progress in Organic Coatings, 2023, 183.

[3]          Different Forms of Corrosion Classified on the Basis of Appearence [M]//BARDAL E. Corrosion and Protection. London; Springer London. 2004: 89-191.

[4]          KWOK C T, MAN H C, CHENG F T, et al. Developments in laser-based surface engineering processes: with particular reference to protection against cavitation erosion [J]. Surface & Coatings Technology, 2016, 291: 189-204.

[5]          KELLYN BETTS. Why small plastic particles may pose a big [J]. Environmental science & technology, 2008

[6]          YOU LI et al. Research on the Influence of Microplastics on Marine Life [J]. IOP Conference Series: Earth and Environmental Science, 2021

[7]          ZHAO X, QI Y, ZHANG Z. Study on anti-cavitation performance of elastomer composite coatings in a salt solution [J]. Progress in Organic Coatings, 2022, 163.

Reviewer 2 Report

Dear Authors,

the paper is a nice summary of the aspects of present combined anticorrosion/antifouling approaches used in marine industry for propeller protection. The subject is important and the review has a good balanced presentation of the background.

On the other hand, when the cited references are looked closely, at cases there are major deviations between the text and the reference related.

After finding some disturbing examples, I took the time and scrolled over ALL the references and checked one by one, wherever it was possible in the internet, if the content of the reference was approriately related to the text of the manuscript. I have found many deviations. I attach a list in a separate document to help the authours to correct their work.

Red letter comments in the attached documents indicate where corrections are definitely needed.

Copyrights have to be checked, in some cases there are missing indications.

English is good.  32. row: spelling mistake "sta te"

Author Response

Response to Reviewer 2 Comments

Point: 6. Yebra, D.M.; Kiil, S.; Dam-johansen, K. Antifouling technology—past, present and future steps towards efficient and environmentally friendly antifouling coatings. Prog. Org. Coat. 2004, 50(2): 75-104.

The data in (6) is also cited in (6) taken from C.D. Anderson, J.E. Hunter, NAV2000 Conference Proceedings, Venice, September 2000.

Response: The reference 6 has been changed to: Anderson, C.D.; Hunter, J.E. International Conference on Ship and Shipping Research; NAV2000 Conference Proceedings, Venice, Italy, 19-22 September 2000.

Point: 7. Tasso, M.; Conlan, S.L.; Clare, A.S. et al. Active Enzyme Nanocoatings Affect Settlement of Balanus amphitrite Barnacle Cyprids. Adv. Funct. Mater. 2012, 22.

This reference is about enzyme based nanocoating and NOT macro organism occurence along coastal area of China as in the manuscript cited

Response: The reference 7 has been deleted.

Point: 9. Dobretsov, S.; Teplitski, M.; Paul, V. Mini-review: quorum sensing in the marine environment and its relationship to biofouling. Biofouling. 2009, 25(5): 413-27.

This paper is about biosignalling and its effect on biofilm dynamics and NOT related to corrosion cases as it is referred in the manuscript.

Response: The reference 9 has been deleted.

Point: 10. Schultz, M.P. Effects of coating roughness and biofouling on ship resistance and powering. Biofouling. 2007, 23(5): 331-41.

This paper does NOT contain the word „corrosion”, so it should not be cited as reference for corrosion cases.

Response: The reference 10 has been deleted.

Point: 17. Patankar, N.A. Transition between superhydrophobic states on rough surfaces. Langmuir. 2004, 20(17): 7097-102.

The paper is about energetics of superhydrophobic surfaces, NOT about fourth stage of biofilm formation, such as macrofouling.

Response: The reference 17 has been deleted.

Point: 19. Liu, D.; Shu, H.; Zhou, J. et al. Research Progress on New Environmentally Friendly Antifouling Coatings in Marine Settings: A Review. Biomimetics. 2023, 8(2).

possible need of copyright?

Response: The reference 19 is in journal “Biomimetics”, which is an open access journal from MDPI.

Point: 20. Plesset, M.S.; Chapman, R.B. Collapse of an initially spherical vapour cavity in the neighbourhood of a solid boundary. J. Fluid Mech. 2006, 47(2): 283-90.

The date 1987 is surely not appropriate... I believe the Authors thought of the 1887 published book of S.W. Barnaby ( https://www.abebooks.com/Marine-Propellers-Being-Course-ThreeLectures/31060593893/bd). This has to be cited as, in fact reference 20 is not even mentioning Barnaby's name!

Response: The reference 20 has been changed to: Barnaby, S.W. Marine Propellers; Being a Course of Three Lectures delivered at the Royal Naval College.; E. & F.N. Spon: London, British, 1887.

Point: 32. Francolini, I.; Vuotto, C.; Piozzi. A. et al. Antifouling and antimicrobial biomaterials: an overview. APMIS. 2017, 125: 392 - 417.

This review is focusing exclusively on coatings on medical devices, for this reason this should not be inserted as such.

Response: The reference 32 has been deleted.

Point: 36. Ye, J.Z.; Chen, S.S.; Ma, C.F. et al. Development of Novel Environment-friendly Antifouling Materials. Surf. Technol. 2017, 46(12): 62-70.

The journal „Surface Technology appeared in Elsevier between 1976 and 1985. In a 2017 release there should be problem.

Response: The reference 36 is in journal“表面工程” not Elsevier, it’s a Chinese article.

Point: 48. Wood, C.D.; Truby, K.; Stein, J. et al. Temporal and spatial variations in macrofouling of silicone fouling‐release coatings. Biofouling. 2000, 16(2-4): 311-22.

This reference would suit better together with referencewith 49 on silicone coating, it has no specialreference to the sentence where it is cited.

Response: According to your opinion, the reference 48 is together with reference 49.

Point: 52. Brady, R.F.J.; Singer, I.L. Mechanical factors favoring release from fouling release coatings. Biofouling. 2000, 15: 73 - 81.

This reference is about mechanical factors and not molecular structure choice as cites.

Response: The reference 52 has been deleted.

Point: 56. Martinelli, E.; Agostini, S.; Galli, G. et al. Nanostructured films of amphiphilic fluorinated block copolymers for fouling release application. Langmuir. 2008, 24(22): 13138-47.

The first athor name is Elisa Martinelli,, so the reference is Martellini et al. and not Elisa et al.

Response: The reference 56 has been changed to Marteinelli et al.

Point: 57. Maan, A.M.C.; Hofman, A.H.; Vos, W.M. et al. Recent Developments and Practical Feasibility of Polymer‐Based Antifouling Coatings. Adv. Funct. Mater. 2020, 30.

Is there a copyright ?

Response: The reference 57 is an open access article, so it doesn’t need copyright. As shown in the figure below.

Point: 70. Ionov, L.; Synytska, A.; Kaul, E. et al. Protein-Resistant Polymer Coatings Based on SurfaceAdsorbed Poly(aminoethyl meth-acrylate)/Poly(ethylene glycol) Copolymers.

Biomacromolecules. 2010, 11(1): 233-7.

The first author name is Leonid Ionov,, so the reference is Ionov et al. and not Leonid et al.

Response: The reference 70 has been changed to Ionov et al.

Point: 73. Blake, J.R.; Leppinen, D.M.; Wang, Q. Cavitation and bubble dynamics: the Kelvin impulse and its applications. Interface Focus. 2015, 5(5): 20150017.

In this paper the authors are „.. developing the Kelvin impulse from first principles, using it, not only as a check on advanced computations (for which it was first used!), but also to provide greater physical insights into cavitation bubble dynamics near boundaries (rigid, potential free surface, twofluid interface, flexible surface and axisymmetric stagnation point flow) and to provide predictions on different types of bubble collapse behaviour, later compared against experiments.” It is not a review and is not about mitigation strategies as it is indicated in the manuscript.

Response: The reference 73 has been deleted.

Point: 74. Chi, S.; Park, J.; Shon, M. Study on cavitation erosion resistance and surface topologies of various coating materials used in shipbuilding industry. J. Ind. Eng. Chem. 2015, 26: 384-9.

This reference is not about surface nitriding technology as indicated in the manuscript.

Response: The reference 74 has been deleted.

Point: 77. Li, Z.; Shen, Y.; Zheng, C. et al. Preparation and properties of fluorosilicone polyether polyurethane underwater acoustically transparent encapsulant. Mater. Today Commun. 2019, 19: 402-6.

In refree’s view this application is not related to propellers

Response: The reference 77 has been deleted.

Point: 85. Qiao, X.N. Preparation and Cavitation Resistance of Polyurethane Elastomer Coatings. Master’s Thesis, Harbin Engineering 955 University, Harbin, China, 2019.

It is mistyped as (75) int he manuscript

Response: The reference 85 has been deleted, and the reference 75 here is correct.

Point: 89. Marlin, P.; Chahine, G. L. Erosion and heating of polyurea under cavitating jets [J]. Wear. 2018.

The first athor name is Pauline Marlin, so the reference is Marlin et al. and not Paulin et al.

Response: The reference 89 has been changed to Marlin et al.

Point: 90. Bonacorso, N.G.; Gonçalves, A.; Dutra, J.C. Automation of the processes of surface measurement and of deposition by welding for the recovery of rotors of large-scale hydraulic turbines. J Mater Process Technol. 2006, 179: 231-8.

In this reference an „ automatic weld filling strategy was designed, implemented and tested” and not inorganic coatings summarized as indicated in the manuscript.

Response: The reference 90 has been deleted.

Point: 102. Huang, W.H.; Chen, K.C.; He, J.L. A study on the cavitation resistance of ion-nitrided steel. Wear. 2002, 252: 459-66

OK – but first author should be mentioned in the manuscript Huang et al.

Response: The reference 102 has been changed to Huang et al.

Point: 107. Yang, H.C. Preparation of Polyolefin Polyurethane Coatings and Their Antifouling and Anticavitation Erosion Performance. Ph.D. Thesis, Harbin Engineering University, Harbin, China, 2021.

The refree has doubts that the economic investigation referenced in the manuscript was coming originally from this thesis. Original reference should be cited.

Response: The economic investigation referenced comes from ETCHNAVIO and EXACTITUDE CONSULTANCY (as shown in the figure below). The reference 107 is to explain that the global imperative for the development of anti-fouling and anti-cavitation coatings is evident, now has been deleted.

Point: English is good.  32. row: spelling mistake "sta te"

Response: We are so sorry for our careless mistakes. We sincerely thanks for your reminder.

Round 2

Reviewer 1 Report

Please once again carefully check and revise the differences between corrosion (driven by physical and/or chemical forces) and wear (to which erosion belongs, and which is driven by mechanical loads), as per definition.

Erosion is not a type of corrosion. But both may occur simultaneously and interact or even enhance their effect on degradation of material.

Thus, simple cavitation erosion in a liquid not acting corrosively is the result of cavitation occuring in the fluid near a solid surface, e.g. if a propeller rotates in water. In the case of additional corrosive attack, e.g. in seawater, there may be a joint action, which is called erosion-corrosion and can be named as a type of tribocorrosion.

Has been improved.

Author Response

Response to Reviewer 1 Comments

Point: Please once again carefully check and revise the differences between corrosion (driven by physical and/or chemical forces) and wear (to which erosion belongs, and which is driven by mechanical loads), as per definition.

Erosion is not a type of corrosion. But both may occur simultaneously and interact or even enhance their effect on degradation of material.

Thus, simple cavitation erosion in a liquid not acting corrosively is the result of cavitation occuring in the fluid near a solid surface, e.g. if a propeller rotates in water. In the case of additional corrosive attack, e.g. in seawater, there may be a joint action, which is called erosion-corrosion and can be named as a type of tribocorrosion.

Response: Thanks for your careful checks. We are sorry for our careless mistakes in the cover letter. Based on your comments, we have made some changes in the revised manuscript and supplemented extra information below.

Erosion is the process by which the action of wind, water, or ice wears away rocks and other materials. Over time, erosion can cause mountains to become valleys and can create new landforms such as canyons and rivers.

The effects of erosion depend on the type of material that is being eroded. For example, if a rock is eroded by wind, the rock will eventually be reduced to sand. If a metal is being corroded by water, the metal will finally be weakened and may break apart.

Corrosion is the process by which metals are slowly eaten away by chemical reactions. This can be caused by air, water, or other chemicals exposure. Over time, corrosion can weaken metal structures and lead to their failure.

The effects of corrosion depend on the type of metal that is being corroded. For example, iron will rust when it comes into contact with water and oxygen, while aluminum will form a protective oxide layer that prevents further corrosion.

Difference Between Erosion and Corrosion:

Difference Between Erosion and Corrosion

Erosion

Corrosion

Erosion is a physical process.

Corrosion is a chemical process.

Occurs on the surface of the land.

Occurs on the surface of materials like polymers, ceramics or metals.

Natural agents like water, gravity, wind, causes erosion.

Corrosive agents such as oxygen, sulfates can cause corrosion.

Erosion involves different processes like transportation, weathering, and dissolution.

Corrosion types include pitting, galvanic, crevice, intergranular and selective leaching.

Land reform techniques like terracing or planting trees can prevent erosion.

The preventive measure includes applying a protective layer on the surface of the metals.

Similarities Between Erosion and Corrosion

Both erosion and corrosion can cause materials to be slowly worn away over time. Additionally, both processes can lead to the failure of structures made from those materials.

Tribocorrosion is a material degradation process due to the combined effect of corrosion and wear. The name tribocorrosion expresses the underlying disciplines of tribology and corrosion. Tribology is concerned with the study of friction, lubrication and wear and corrosion is concerned with the chemical and electrochemical interactions between a material, normally a metal, and its environment. Tribocorrosion concerns the irreversible transformation of materials or of their function as a result of simultaneous mechanical and chemical/electrochemical interactions between surfaces in relative motion.

Wear is a mechanical material degradation process occurring on rubbing or impacting surfaces, while corrosion involves chemical or electrochemical reactions of the material. Corrosion may accelerate wear and wear may accelerate corrosion. One then speaks of corrosion accelerated wear or wear accelerated corrosion. Both these phenomena, as well as fretting corrosion (which results from small amplitude oscillations between contacting surfaces) fall into the broader category of tribocorrosion. Erosion-corrosion is another tribocorrosion phenomenon involving mechanical and chemical effects: impacting particles or fluids erode a solid surface by abrasion, chipping or fatigue while simultaneously the surface corrodes.

And propeller cavitation in real marine environment involves the cooperative effects of mechanical-related corrosion and electrochemical corrosion, which act synergistically to accelerate the corrosion processes, therefor can be called erosion-corro
